# Lack of Efficacy of Albendazole against *Dicrocoelium dendriticum* Infection in a Sheep Farm in France

**DOI:** 10.3390/ani14131992

**Published:** 2024-07-05

**Authors:** Julie Petermann, Christelle Grisez, Sophie Lavigne, Philippe Jacquiet

**Affiliations:** 1IHAP 1225, Université de Toulouse, INRAE ENVT, 31076 Toulouse, Francephilippe.jacquiet@envt.fr (P.J.); 2CIIRPO, Ferme du Mourier, 87800 Saint-Priest-Ligoure, France

**Keywords:** *Dicrocoelium dendriticum*, albendazole, lack of efficacy, sheep

## Abstract

**Simple Summary:**

Infection by the parasite *Dicrocoelium dendriticum* has an impact on the production and health of sheep. Because the parasite cycle is complex, management is based on antiparasitic treatment. Albendazole at a dose of 15 mg/kg is known to be effective against the adult stages of *D. dendriticum.* Following a suspected lack of efficacy of this treatment in a French sheep farm, we carried out an efficacy test by treating 15 ewes at a dose of 15 mg/kg. We obtained a treatment efficacy of 39%, well below the 90% efficacy or more observed in other studies with the same compound at the same dosage. This lack of efficacy of albendazole on this farm is a warning for sheep farms, as there are no other active molecules against *D. dendriticum* currently available in France.

**Abstract:**

Dicrocoeliosis is a common parasitic disease in European sheep farming. The prevalence of infection by this parasite can reach almost 70% in areas where the environment is favorable to intermediate hosts. In France, only one drug is currently available for the treatment of dicrocoeliosis: albendazole at a dose of 15 mg/kg in a single administration. However, a control coproscopy following a routine treatment led us to suspect that the efficacy of albendazole against *Dicrocoelium dendriticum* had diminished. Therefore, we carried out an efficacy test on 15 animals by treating them with albendazole at a dose of 15 mg/kg and performing a coproscopy on D0 and a control coproscopy 14 days later. We obtained a 39% reduction in the excretion of *D. dendriticum* eggs. This shows a reduction in the expected efficacy of albendazole, which is normally more than 90% in other studies involving this molecule at a dosage of 15 mg/kg. These results are of major concern as albendazole is currently the only drug available in France to treat dicrocoeliosis.

## 1. Introduction

Dicrocoeliosis linked to the parasite *Dicrocœlium dendriticum* is a common parasitic disease of ruminants in Europe. A study by Cringoli et al. showed a prevalence of infection of 67.5% in sheep during a study in the southern Italian Apennines [1], while in Sardaigne, it was 51.1% [2]. Similarly, a study carried out in the province of Leon, in north-west Spain, showed a prevalence of 63.6% in sheep [3]. These various studies show that *D. dendriticum* is a parasite frequently found in these regions of southern Europe.

*D. dendriticum* is a parasite that lives in the bile ducts and gall bladder [4]. The clinical signs associated with infection by *D. dendriticum* are difficult to characterize, as it is most often a case of subclinical infection and a combination of different parasite species [4,5]. As adults, *D. dendriticum* are located in the bile ducts and gall bladder; the impact often involves hepatic disturbances with increases in liver enzymes, in particular alanine aminotransferase (ALT) and aspartate aminotransferase (AST), regularly demonstrated during natural or experimental infections [6,7]. However, there does not appear to be any drop in hematocrit or plasma albumin when infection levels at necropsy are below 4000 flukes [8]. In some cases, treatment leads to clinical improvement with a reduction in anemia and jaundice and an improvement in milk production or body condition score, but these are natural infections associated with multiple parasites, so it is difficult to impute the clinical signs to *D. dendriticum* alone [7,9]. However, a significant difference in weight was observed in experimentally infected lambs compared with healthy animals [6].

*D. dendriticum* is a parasite with a complex cycle involving two intermediate hosts, a snail and an ant. Several species of terrestrial gastropods and ants can serve as intermediate hosts for the *D. dendriticum* cycle [10]. The prevalence of small fluke infection in sheep is linked to the presence of these intermediate hosts. The prepatent period is 49 to 79 days under experimental infection conditions [11]. Excretion is regular throughout the year, with a slight peak in autumn and winter [3]. Excretion appears to be positively correlated with the number of adult worms present in the animal [11].

In general, the diagnosis is made by a post-mortem examination focusing on the gall bladder and bile ducts. This enables the adult parasites to be identified directly. Diagnosis can also be made while the animal is still alive by analyzing the fecal excretion of *D. dendriticum* eggs. Analysis by flotation requires dense flotation liquid (>1.33) [5,12,13]. Fecal examination seem to be a good diagnostic tool as it appears to be a positive association between the number of adult worms in the liver and egg excretion in the animals’ feces [11,14,15]. Moreover, Pieragi has also demonstrated a positive association between the number of adult worms and hepatic lesions such as hepatic fibrosis and bile duct hyperplasia [14].

Concerning treatment, several compounds have been tested to combat dicrocoeliosis. Albendazole is the molecule most frequently tested for the treatment of *D. dendriticum* [9,16,17,18]. This molecule shows efficacy ranging from 92 to 96% reduction in fecal excretion when administered orally at a single dose of 15 mg/kg and 10 mg/kg [17,18], as well as when administered repeatedly at 7-day intervals (99% reduction, [17] or when administered as a sustained release bolus (88.5% to 91.8% reduction in excretion, [9,16]). However, albendazole at a dose of 5 mg/kg appears insufficient to control dicrocoeliosis [19]. Other compounds like thiophanate and netobimin have also been shown to be effective against small flukes. Netobimin at a dose of 15 mg/kg or 20 mg/kg showed efficacies of 91.9% and 91.5%, respectively [20]. This study was carried out to compare the efficacy of two suspensions (5% and 15%) of netobimin against *D. dendriticum* in naturally infected sheep. The efficacy was checked by post-mortem examination twenty-one days post-treatment. The results showed that the efficacies of 5% and 15% suspension of netobimin were 90.80% and 91.50%, respectively, and the use of 15% suspension in sheep is discussed in [20,21]. Thiophanate showed 74.4% efficacy at a dose of 50 mg/kg [19]. A molecule from another family, praziquantel, has also been tested on infected sheep and llamas at a dose of 50mg/kg and appears to be effective [22,23].

The fact that led to our study was a fecal egg count carried out after albendazole treatment at a dose of 15 mg/kg. Following this treatment, a coproscopic analysis was carried out on a composite sample to check the parasitic status of the animals. Small fluke eggs were found, leading us to suspect a lack of efficacy of albendazole on this farm. To verify this hypothesis, we therefore decided to test the efficacy of albendazole on this farm more specifically.

## 2. Materials and Methods

### 2.1. Farm Presentation

The study took place on Le Mourier research and innovation site, which is a sheep farm in the Limousin region in the center of France. In this geographical area, the climate is oceanic. The farm, located at an altitude of 300 m, has 50% permanent grassland and 50% temporary grassland. This sheep farm has 700 ewes, half of which are of the Vendéenne breed and the other half F1 (Île de France crossed with Romanov).

This farm is used to treat against *D. dendriticum* and a fecal examination after a usual treatment leads to a suspicion of lack of efficacy of albendazole against *D. dendriticum* at 15 mg/kg.

### 2.2. Study Design and Efficacy Calculation

Sampling and analysis were carried out in accordance with the recommendations of the World Association for the Advancement of Veterinary Parasitology WAAVP [24].

The animals included in the study were in a group of 60 ewes known to be infected with *D. dendriticum*. Ewes from this group were taken at random, representing each age group. There were 15 ewes of the Vendéenne breed, in a very good body condition (body condition score above 4) and not pregnant. A fecal sample was taken directly from the rectum of these ewes on 13 February 2024 for coprological analysis. The same day, the ewes were weighted individually and orally treated with 15 mg/kg albendazole Valbazen^®^ (Valbazen moutons et chèvres 1.9%, Zoetis, France) in accordance with the marketing authorization recommendations. For each ewe, the dosage was adapted to the body weight of the ewe, and the treatment was administered with a syringe in order to achieve a very precise dosage (Table 1).

The ewes had been grazing for over three months prior to this treatment and remained on pasture for the duration of the experiment. Fecal samples were taken again on 27 February 2024.

### 2.3. Fecal Examination

Samples were stored at room temperature and analyzed within 24 h of collection.

They were analyzed using the McMaster technique modified by Raynaud [25]. A total of 1 g of fecal material was diluted in 14 mL of flotation liquid with a specific density of 1.45 (LST FASTFLOAT—Pangea UK), with a sensitivity of 15 eggs per gram. The individual egg count is an interesting alternative to autopsy as there is a positive correlation between the number of eggs found in feces and the number of adult worms found in the liver [11,14,15].

Individual data on fecal egg counts before and after treatment were analyzed in the web application (https://www.fecrt.com, accessed on 5 March 2024). Upper and lower 90% confidence intervals (CIs) for the percentage reduction in fecal egg count two weeks post-treatment were calculated using the hybrid Bayesian-frequentist inference method (pairwise study design; research protocol [26]).

A Friedman test was realized to evaluate the differences in fecal excretions between the day of treatment and 14 days after treatment.

Feces collection and anthelmintic treatments are part of routine veterinary procedures without any traumatic methods, so no specific ethical authorization was required.

## 3. Results

All fifteen animals excreted *D. dendriticum* eggs in their feces at D0 and D14 after treatment (Table 1). The excretion values at D0 and D14 are shown in Figure 1. The mean excretion at D0 was 347 eggs per gram with a minimum of 100 epg and a maximum of 1000 epg. At D14, the mean excretion was 211 epg with a minimum of 30 epg and a maximum of 550 epg. There was no significant decrease in egg excretions between the day of treatment and 14 days after treatment (*p* = 0.1967, Friedman test).

The reduction in fecal egg excretion between D0 and D14 was 39% with a 90% confidence interval calculated according to the methods recommended by the WAAVP [12] ranging from −1.1% to 63.2%.

The result of the efficacy test, therefore, shows that albendazole lacks efficacy against *D. dendriticum* under our experimental conditions.

## 4. Discussion

To date, there has never been a description of the lack of efficacy of albendazole against small flukes [27], and a recent study re-confirmed the efficacy of albendazole at 15 mg/kg in a single oral administration against *D. dendriticum* [18]. This is the first time that an efficacy of much less than 90% has been observed. Previously published studies have reported an efficacy above 90% with this molecule and dosage [17,18].

Albendazole is not effective on larvae [28]. The lack of efficacy observed in our study could be linked in part to the evolution of larval stages into adult stages. However, in studies showing albendazole to be more than 90% effective, the same phenomenon can be observed, especially with Gortva dairy sheep in the Konigova study in which ewes grazed until the beginning of the experiment [18]. It would therefore seem that this phenomenon could explain an efficacy below 100%, but not as collapsed as in our present study.

This initial finding of the reduced efficacy of albendazole raises an important issue in sheep farming. The mechanism of action of albendazole is to act on microtubules’ polymerization [29]. In other parasite species where the mechanism of action is similar, resistance to one molecule in the family leads to resistance to the whole benzimidazole family [30]. A similar reaction is to be feared for *D. dendriticum*. This would mean that the lack of efficacy is likely to be present for both molecules: albendazole and netobimin. Moreover, at present, thiophanate and netobimin are no longer available in France. This raises the question of the possibility of using an alternative treatment for dicrocoeliosis.

It might be interesting to test praziquantel for the treatment of small flukes, this being the only molecule outside the benzimidazole and pro-benzimidazole family [22,23]. The problem is that the dose that seems to be effective corresponds to almost 14 times that used to treat *Moniezia expansa* infections according to the manufacturer’s recommendations [22,23]. Another molecule that can be tested is triclabendazole. Although it is a molecule of the benzimidazole family, the mode of action is different [31]. This molecule has been used successfully in humans to treat dicrocoeliosis at 10 mg/kg in a single take [32,33].

These results also show the importance of verifying the efficacy of treatments by post-treatment fecal egg counts. Since the signs of dicrocoeliosis are subclinical and not pathognomonic, it is very difficult to determine whether treatment has been ineffective. This is even more true given that parasitic infections are generally multiple, and the imputability of *D. dendriticum* is tricky [5,19]. Furthermore, testing for *D. dendriticum* requires a dense flotation fluid and is therefore not routinely performed [13].

Dicrocoeliosis is a seldom-studied disease in France. One study on goats carried out in a limited region showed a prevalence of 20% [34]. This prevalence is higher in other parts of Europe. Alkaline soils are favorable to *D. dendriticum* infestation because they are conducive to intermediate hosts. Although we have good knowledge of the parasite cycle [10], we have little information on the epidemiology of the disease in France. It would be interesting to improve this knowledge in order to make better recommendations to farmers.

The result of our study showing a reduction in the efficacy of albendazole is problematic because there is no alternative means of management. In fact, it is unthinkable to control intermediate hosts for reasons of ecological impact and cost. The only recommendations that can be made are to avoid total grazing in the morning and evening to limit contact with ants, but this is unthinkable in systems where grazing is continuous over an entire period and the ewes do not return to the barn at night [5].

With the increase in resistance among strongyle species [35,36], mixed grazing is becoming a common recommendation in order to control strongyle population. However, *D. dendriticum* should be kept in mind when grazing occurs with multiple species, especially when it involves herds of different origins. Mixing-grazing herds may be at risk of spreading resistance.

Finally, this first identification of the lack of efficacy of albendazole highlights the fact that post-treatment fecal egg count should be used in order to check the efficacy of the treatment. It is important that sheep breeders with a lack of efficacy of treatment against *D. dendriticum* stop spending time or money using treatments that are not sufficiently efficient. This highlights also the need for alternative treatment for such parasitic infections.

## 5. Conclusions

This is the first report on the lack of efficacy of albendazole against sheep infection with *D. dendriticum*. Although albendazole is not effective against the larval stages of small flukes, the reported efficacy of albendazole at 15 mg/kg is generally higher than 90%. The lack of efficacy of albendazole highlights the importance of preserving the efficacy of these molecules by using them only when necessary, thus improving the diagnosis of dicrocoeliosis. It is also becoming important to regularly check the effectiveness of treatments by post-treatment fecal analysis. As well as improving diagnosis, it is becoming necessary to find alternative methods to control *D. dendriticum*. This requires a thorough understanding of the parasite, the epidemiology of infections, and the clinical impact in order to identify new management measures.

## Figures and Tables

**Figure 1 animals-14-01992-f001:**
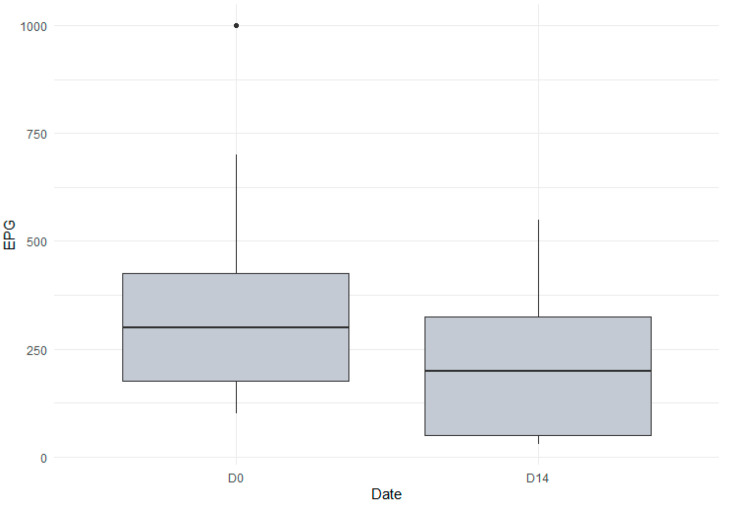
Fecal egg counts of *Dicrocoelium dendriticum* on day of treatment and 14 days after treatment with albendazole.

**Table 1 animals-14-01992-t001:** Ewes weight, dose of albendazole, and individual fecal egg excretions on day of treatment and 14 days after treatment with albendazole.

Ewes Number	D0			D14
	Ewes Weight (kg)	Volume of Drug (Valbazen Moutons et Chèvres 1.9%, Zoetis, France) Administered (ml) Corresponding to a Dose of 15 mg/kg of Body Weight	EPG of *D. dendriticum* Day of Treatment	EPG of *D. dendriticum* 14 Days after Treatment
1	73	60	250	450
2	82.2	72	200	50
3	77.2	64	300	550
4	81.8	72	350	60
5	80.6	64	700	500
6	71.3	60	500	300
7	66.4	56	250	30
8	76.2	64	100	200
9	66.8	56	450	50
10	67.4	56	1000	350
11	54.1	44	100	200
12	57.6	48	300	30
13	76.4	64	150	100
14	86.9	72	150	250
15	97.8	80	400	50

## Data Availability

All the data are available in the Table 1 of the article.

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
