# Peer review of "Lack of Efficacy of Albendazole against Dicrocoelium dendriticum Infection in a Sheep Farm in France"

_animals, 2024, doi:10.3390/ani14131992_

Round 1
Reviewer 1 Report
Comments and Suggestions for Authors
The work offers important insight into the first report of possible resistance, however a couple of questions need to be improved
1) In the section Study design, efficacy calculation
Please add a subsection specifying more about the animals used. The information is scattered throughout the paragraphs and makes reading difficult.
The authors mention that for the study group, a fecal sample was taken and on the same day they were treated with albendazole.
My question is, did the authors have the results of the stool analysis before performing the treatment? probably not, since in the fecal examination section it says that the samples were processed within 24 hours, so it is not clear how the first treatment was carried out.
Please clarify this and rewrite the information.
2) In the disscusion section
The authors mention that it could be interesting to try praziquantel for the treatment of small flukes, this being the only molecule outside the benzimidazole and probenzimidazole family and you mention some considerations regarding the dose.
In this sense, there are previous reports of infections in humans successfully treated with triclabendazole, which, although it is a bendimidazole drug, does not share the same mechanism of action as those mentioned above. In fact, it is widely used as a fasciolicide, both in humans and animals.
It would be good to add a paragraph about that in the discussion.
Author Response
Dear Reviewer 1,
Please find below the informations concerning the corrections requested :
1) In the section Study design, efficacy calculation
Please add a subsection specifying more about the animals used. The information is scattered throughout the paragraphs and makes reading difficult.
The authors mention that for the study group, a faecal sample was taken and on the same day they were treated with albendazole.
My question is, did the authors have the results of the stool analysis before performing the treatment? probably not, since in the fecal examination section it says that the samples were processed within 24 hours, so it is not clear how the first treatment was carried out.
- See Table 1
2) In the disscusion section
The authors mention that it could be interesting to try praziquantel for the treatment of small flukes, this being the only molecule outside the benzimidazole and probenzimidazole family and you mention some considerations regarding the dose.
In this sense, there are previous reports of infections in humans successfully treated with triclabendazole, which, although it is a bendimidazole drug, does not share the same mechanism of action as those mentioned above. In fact, it is widely used as a fasciolicide, both in humans and animals.
It would be good to add a paragraph about that in the discussion.
- Done L 177-179
Reviewer 2 Report
Comments and Suggestions for Authors
Reviewer’s comments and suggestions:
Infection by the parasite Dicrocoelium dendriticum has an impact on the production and health of sheep in Europe. Albendazole as the effective drug, it is valuable to evaluate its efficacy. Significantly, the research should be scientific and rigorous. There are some problems should be improved.The specific comments and suggestions are as follow.
1. Abstract. It is necessary to add the significance of this study at the end of this section.
2. Keywords. Please add “Sheep”.
3. Introduction.
(1) There are too many paragraphs in this section, it is suggest to merge suitably. For example, the relevant information such as the detection method of the disease should be written into a statement.
(2) There was no close connection between paragraphs, please revise.
4. Materials and Methods.
(1) Farm presentation. Line 96, what is the meaning of the different sheep breeds mentioned here? Why use Vendéenne breed only for the research? Do they have some advantages for this research?
(2) Study design, efficacy calculation. Because the previous study proved that the albendazole has efficacy, the method in this study obtained the different results, which should be more rigorous and prove the results by bidirectional verification. For example, it is recommended to add an additional group with more doses or use drugs with double double treatment.
(3) Faecal examination. The number of worm eggs can not indicate the infection status of the worms, then it is not enough accurate to perform a faecal examination alone, so it is necessary to supplement the experiments with autopsy of sheep and show the results in this MS.
5. Results. A table is necessary to describe the condition of each sheep, such as EPG and deworming clean rate etc.
6. Conclusion. You are supposed to describe it in brief words.
7. Please check the details carefully and correct.
(1) Latin name should be in full name when it first appears, and use abbreviations when it appears again.
(2) Parasite genus names should also be capitalized and italicized, such as Line66, “dicrocoelium” should be written as“Dicrocoelium”.
(3) Please review and modify the format of references according to the requirement by the Journal. There are many parasite names that do not use italics. And please check the problems with word capitalization in article titles when quoting references.
Comments on the Quality of English Language
Please see the attachment.
Author Response
Dear Reviewer 2,
Please find below the informations concerning the corrections requested :
- Abstract. It is necessary to add the significance of this study at the end of this section.
=> Done l 25-26
- Keywords. Please add “Sheep”.
=> done l 28
- Introduction. => done
(1) There are too many paragraphs in this section, it is suggest to merge suitably. For example, the relevant information such as the detection method of the disease should be written into a statement.
(2) There was no close connection between paragraphs, please revise.
- Materials and Methods.
(1) Farm presentation. Line 96, what is the meaning of the different sheep breeds mentioned here? Why use Vendéenne breed only for the research? Do they have some advantages for this research?
=> The Vendéenne breed didn’t show particularly advantage except the fact that we knew they were significantly infested due to previous analyse for this group of ewes. I put a small paragraph to explain that part l 102-104
(2) Study design, efficacy calculation. Because the previous study proved that the albendazole has efficacy, the method in this study obtained the different results, which should be more rigorous and prove the results by bidirectional verification. For example, it is recommended to add an additional group with more doses or use drugs with double double treatment.
=> In fact, we wanted to test if there were still some reduction in eggs excretion with albendazole used at the dose recommend in the marketing authorization, as it is used in farms.
=> The aim of our study is to alert veterinarian and breeder on the fact that sometimes albendazole is not as effective as we intended
=> I had the p-value between the excretion at day 0 and day 14. L128/138
(3) Feacal examination. The number of worm eggs can not indicate the infection status of the worms, then it is not enough accurate to perform a faecal examination alone, so it is necessary to supplement the experiments with autopsy of sheep and show the results in this MS.
=> It should be interesting to do autopsy to confirm the result observed but it is difficult to do autopsy in farm, especially when ewes are in quiet good health and conditions. Moreover, as there is a positive correlation between eggs count and the number of worms, we think that this is a good tool to evaluate the presence of adult worms. In fact, if there is eggs, there are adults worm. The only case in which we can still find eggs is if they are stored in the gall bladder but in fact it’s a very few eggs (Flanagan et al., 2011)
=> I had information about positive correlation L119-122
- Results. A table is necessary to describe the condition of each sheep, such as EPG and deworming clean rate etc.
=> ok table 1
- Conclusion. You are supposed to describe it in brief words.
=>Sorry, but I don’t really understand the demand. May you precise please?
- Please check the details carefully and correct.
(1) Latin name should be in full name when it first appears, and use abbreviations when it appears again.
=> Done
(2) Parasite genus names should also be capitalized and italicized, such as Line66, “dicrocoelium” should be written as“Dicrocoelium”.
=> Done
(3) Please review and modify the format of references according to the requirement by the Journal. There are many parasite names that do not use italics. And please check the problems with word capitalization in article titles when quoting references.
=> Done
Round 2
Reviewer 2 Report
Comments and Suggestions for Authors
The authors have addressed the comments and suggestions of reviewer, appropriately explained our questions, and the revised manuscript is acceptable for publication.